# Hand Erosive Osteoarthritis and Distal Interphalangeal Involvement in Psoriatic Arthritis: The Place of Conservative Therapy

**DOI:** 10.3390/jcm10122630

**Published:** 2021-06-15

**Authors:** Elena Poletto, Ilaria Tinazzi, Antonio Marchetta, Nicola Smania, Elena Rossato

**Affiliations:** 1Neuromotor and Cognitive Rehabilitation Research Center, Physical and Rehabilitation Medicine Section, Department of Neurosciences, Biomedicine and Movement Sciences, University of Verona, 37129 Verona, Italy; elena.poletto@hotmail.it (E.P.); nicola.smania@univr.it (N.S.); 2Rheumatology Unit, IRCCS Sacro Cuore Don Calabria, 37024 Negrar, Italy; ilaria.tinazzi@sacrocuore.it (I.T.); antonio.marchetta@sacrocuore.it (A.M.); 3Rehabilitation Department, IRCCS Sacro Cuore Don Calabria, 37024 Negrar, Italy

**Keywords:** erosive hand osteoarthritis, psoriatic arthritis, conservative therapy

## Abstract

Hand erosive osteoarthritis (HEOA) and Psoriatic Arthritis (PsA) with DIP involvement are common diseases affecting the hand. Both of them evolve with a progressive limitation in grip due to limited range of motion of the affected joints and stenosing tenosynovitis. Pharmacological options currently available (corticosteroids and clodronate or Idrossicloroquine) for the treatment of EHOA are mostly symptomatic and currently there are no effective drugs able to modify the course of the disease. In addition, data on drug effectiveness of PsA with DIP involvement are lacking. Conservative therapy should be considered in order to reduce pain and improve hand functionality. There are many studies debating a wide range of non-pharmacological intervention in the management of HEOA: joint protection program, range of motion and strengthening exercise, hand exercise with electromagnetic therapy, application of heat with paraffin wax or balneotherapy, occupational therapy and education. Concerning conservative treatment strategies to treat PsA, on the contrary, current evidence is still weak. Further research is needed to find the correct place of physical therapy to prevent stiffness and ankylosis due to the vicious circle of inflammation-pain-immobility-rigidity.

## 1. Background

Among inflammatory arthropathies, one of the biggest challenges is considered to be differentiating between the diagnosis of psoriatic arthritis and erosive osteoarthritis [1,2], as both conditions preferentially affect distal IP joints with similar clinical features.

Osteoarthritis (OA) is the most common joint disease in the world and one of the main causes of long-term disability in people over 65 yrs.

Erosive osteoarthritis is an uncommon variant of OA (EHOA) defined by Peter et al. and characterized by degenerative changes and inflammatory episodes of the interphalangeal joints (IP) [3,4,5] mainly in middle-aged women [6].

According to the criteria of European League against Rheumatism (EULAR), EHOA is defined radiographically by subchondral erosion, cortical destruction and the subsequent reparative changes, such as bony ankylosis [7].

Even though EHOA has clinical evolution that is different from classical OA, the relation between classical OA and EHOA is controversial; even if erosive hand OA is more severe from the onset, some consider EHOA to be an advanced stage of the same process underlying classical OA [8]. However, the core element of EHOA is its higher disease burden with more structural damage as well as inflammation compared with classical form.

In a prevalence study, Carvasin et al. examined 640 subjects aged >40 from the Venetian area. 31% of subjects suffered from hand osteoarthritis according to Altman criteria; 7% of those showed radiological aspects of EHOA [9].

Psoriatic arthritis (PsA), on the contrary, has been defined as a chronic inflammatory arthritis, associated with psoriasis, which frequently involves enthesis, peripheral joints and the spine [10]. The female-to-male ratio of the PsA patients was close to one.

Up to 30% of patients with psoriasis have arthritic involvement using CASPAR criteria [11,12].

Similar to HEOA, PsA could progress to a very destructive disabling form of distal interphalangeal arthritis [13]. Indeed, bone erosions occur in up to 47% of PsA patients within 2 years of disease onset [14].

### 1.1. Pathogenesis

The pathogenesis of PsA as well as HEOA is multifactorial including biomechanical, metabolic, hormonal and genetic factors [15,16].

Even though osteoarthritis was defined as a disease of articular cartilage, research evidence well established that it involved the entire joint, with changes that occur also in subchondral bone, synovial membrane and ligaments.

Excessive mechanical load, such as obesity, joint misalignment or hyperlaxity and many other factors predispose and promote the imbalance between metabolic and degradative processes in favor of degradation [15]. Proinflammatory mediators, including cytokines, chemokines and proteolytic enzymes participate from the earliest phases of OA and are associated with early damage.

### 1.2. Clinical Features

Differentiating PsA from EHOA is one of the most intriguing challenges [1,2].

Psoriatic arthritis occurs as an inflammatory manifestation (joint swelling, pain and tenderness, erythema, warmth and functional limitation) of the joints, tendons and ligaments.

EHOA is clinically characterized by abrupt and painful onset, swelling and pain, as well as functional limitation. Episodes of inflammatory flares are recurrent in the course of the disease and justify the term “inflammatory OA” [1,6].

Some peculiar clinical features could help to orientate diagnosis towards EHOA or PsA, especially when PsA involve other sites and tissues in addition to distal interphalangeal joints (Table 1).

In the majority of PsA patients, skin involvement precedes the onset of psoriatic arthritis (PsA) by many years, with a median of 10 [11].

Thus, a prompt diagnosis of PsA can be achieved with an accurate clinical assessment and evaluation of skin and nail lesions in patients with other musculoskeletal disorders [17]. With this purpose, in clinical setting the presence of a multidisciplinary team including dermatology and rheumatology specialists could increase early diagnosis and optimize care of PsA patients [18].

Up to 40% of patients with psoriatic arthritis developed dactylitis, which provides an important clue in diagnosing PsA [19].

Spinal involvement, enthesitis, nail dystrophy and uveitis are other representative features.

When PsA involves exclusively DIP or PsAprecedes skin involvement, the diagnosis becomes more challenging.

Both PsA and EHOA have erosive damage as a core feature; erosion typically affects distal interphalangeal (DIP) and proximal interphalangeal (PIP) joints, with preference for DIP in both conditions.

Symmetric joint distribution is however a remarkable aspect of EHOA while PsA has typically asymmetrical fashion [1,13].

Heberden’s nodes (DIP) and Bouchard’s nodes (PIP) are classical joint deformities of all types of hand osteoarthritis; tough nodose deformities are less frequent in erosive form than in non-erosive form, they can be a useful feature to distinguish EHOA and PsA [1,20]. Medial subluxation of the middle phalanx, flexion deformities and ankylosis are other leading structural damage that occur more frequently in EHOA [6,15,21,22].

EHOA and PsA have considerable functional consequences such as reduced hand mobility and grip force with considerable impact on everyday life autonomies.

EHOA is associated with more severe functional limitations [8]; loss of grip strength is strictly related to erosions [23]. The joint space narrowing, a core feature of EOA and PsA, is strictly related to loss of articular function; the resulting repetitive friction of bone extremity in advanced stages of the disease leads to deformities [24].

Both EHOA and PsA develop severe deformities of interphalangeal joints which reflect on pinch strength and the ability of prehension and lead at least to limitation on several daily functional activities such as writing, handling or fingering small objects. As a consequence, patients often require orthopaedic surgical corrective treatment.

Severe pain and inflamed joints clearly contribute to the degree of functional limitations.

Hulsted et al. demonstrated in PsA that the number of active inflamed Joints is related to progressive physical deterioration [25].

Consequently, to avoid pain, patients decrease joint mobility and therefore perpetrate the vicious circle of pain-immobility-rigidity. Moreover, the average grip strength in women with hand OA was <60% that of healthy age and sex-matched individuals and the reduced grip strength was strictly correlated with activity limitation and participation restrictions [26].

Loss of hand movement and function reduce quality of life of these patients and also increase the risk of developing depressive symptoms [27].

### 1.3. OA Erosive Arthritis and DIP PsA Similar Imaging Features

In case of PsA without skin involvement and exclusive involvement of DIP, clinical findings might not be enough to discriminate between EHOA and PsA. In this case, differential diagnosis relies mostly on radiological features [2].

Indeed, radiography is the gold standard in imaging of osteoarthritis and PsA both for diagnosis and to assess progressive damage.

Both disorders have morphological similarities such as erosions, joint space narrowing and subchondral sclerosis.

In EOA erosion typically begins at the central portion of articulation, sharply marginated.

In PsA, on the other hand, erosions appear first in the marginal part of the joint and extend into the center.

In early phases bone destruction is well-defined, whereas in later stages erosions may become irregular and undefined [13,28].

Marginal sclerosis and osteophytes of the distal plate together with central erosions or collapse and thinning of the proximal phalanx are responsible for the “gull wings” appearance in EHOA [1,29] (Figure 1).

Marginally located erosions in proximal subchondral cortex and marginal periostitis in the distal plate at DIP joints giving a characteristic appearance of “mouse ears” in PsA (Figure 2).

“Saw tooth” appearance is another characteristic configuration of EHOA, found especially in PIP [1] (Figure 1).

Osteolytic lesions of the phalanx in psoriatic arthritis could lead to the so called “pencil in the cup” deformity, the most severe form of telescoping of digits, acro-osteolysis and mutilant arthritis which are peculiar of PsA [2,13,28].

Another important radiological difference between these two conditions concerns pathological new bone formation.

Osteoarthritis has been mainly characterized by a combination of bony proliferation and erosion. In EHOA the presence of osteophytes, linear bone apportion predominantly at the margins of DIP and PIP joint, are the major radiographic aspect of osteoarthritis [6].

The occurrence of new periosteal bone proliferation and cartilage destruction is one of the most distinctive aspect also of PsA. Another one is periostitis with an irregular pattern of apposition: distinctive spike-like or fluffy proliferative alterations surrounded erosions on joint surface margins and phalangeal tuft [2,28].

Bone ankylosis is a core feature of PsA, but it can also occur in HEOA [6,28].

The selective radiologic involvement of a single ray provides a useful pathognomonic clue of psoriatic arthritis as regards the soft tissue swelling of dactylitis which could be modest and not so evident, and it made clinical findings not effective [27].

Accumulating evidence is enhancing the role of ultrasonography as a useful, low-cost non-invasive technique to identify diagnostic features of EHOA and PsA [20,30,31,32,33]. US, in fact, is able to visualize discontinuity of bone surface, joint space narrowing and osteoproliferation. Ultrasonography was demonstrated to be more sensitive in the detection of central erosions of osteoarthritis with respect to plain radiographs [31]. EHOA inflammatory activity is expressed as synovial thickening and excessive intra-articular power Doppler signal with higher sensitivity than clinical assessment [31,32].

The high sensitivity of ultrasound technique to detect inflammatory signals [32,34] made this methodology able to provide detailed information also regarding PsA activity and features, as with US it has been possible to identify synovitis since subclinical stage [33]. Doppler may finally be used to identify specific features of PsA such as enthesitis and dactylitis [33] (Figure 3).

Further imaging modalities provide reliable results in detection of typical lesions of both HEOA and PsA and help to make accurate diagnosis, as proved in clinical trials [35].

As mentioned, differential diagnosis between HEOA and PsA may sometimes become largely dependent on imaging interpretation results. Magnetic resonance imaging (MRI) is a valid operator-independent tool and it has a higher sensitivity than radiography and US in detecting bone erosion, especially in earlier stages of HEOA [36,37]. Moreover, MRI can detect damage in all the joint structures in PsA patients (i.e., cartilage defects, bone marrow oedema, soft tissue oedema and synovitis) [38]. MRI is not mandatory in all patients, but it could support diagnosis when clinical and other radiological findings are not conclusive.

It has been shown that PsA and HEOA could coexist with gout and calcium pyrophosphate deposition disease (CPPD) [2,35]. In particular, considering HEOA, joint damage has a crumbling appearance similar to CPPD and CPP deposition are not so rarely detected in osteoarthritis; thus, differential diagnosis could be difficult. Recently, it has been demonstrated that Dual energy Computerized Tomography (DECT) directly visualizes and quantifies calcium pyrophosphate (CPP) and urate monosodium (MSU) crystal [35]. Moreover, DECT, in contrast with conventional CT, DECT is able to detect inflammatory lesions. Hence in this context DECT seems a valuable tool to achieve a more definite diagnosis.

### 1.4. Lack of Evidence for Effective Systemic Therapy in EHOA

Pharmacological options currently available for the treatment of EHOA are mostly symptomatic with no clear improvement on the progression of the diseases to erosive destructive lesions [39].

Most frequently administered agents in EHOA are acetaminophen, non-steroidal anti-inflammatory drugs (NSAIDs), both topical and oral preparation and cyclooxygenase-II (COX-II) inhibitors [39].

Considering their robust anti-inflammatory effect, glucocorticoids were also investigated as therapeutic option with contrasting results. Treatment of patients with symptomatic HOA which show sign of inflammation with 10 mg of prednisolone per day for 6 weeks resulted in pain relief in a double-blind, placebo-controlled trial [40]. This conclusion was in contrast with a previous literature study using 5 mg of prednisolone for 4 week that was not proven to be superior to placebo [41]. Furthermore, intra-articular injection of glucocorticoid in DIP/PIP joints seems to be effective to decrease pain and inflammation but the effect is strictly confined to the injected joint [42].

The role of Idrossicloroquine (HCQ) in EHOA treatment is also controversial because only few RCTs (randomized controlled trials) investigate it and the results are contradictory besides the potential adverse events [43,44,45,46,47]. Two large European trials investigated the analgesic and anti-inflammatory effects of HCQ: in both multicenter RCTs, HCQ was not able to relieve pain or reduce inflammatory changes [45,46].

Three RCTs have also investigated the efficacy of adalimumab, a tumor necrosis factor (TNF)-α-antagonist, compared to placebo in patients with EHOA. Hence the overall results showed that adalimumab is not superior to placebo to alleviate symptoms or synovitis [48,49,50].

Furthermore, a recent study—The E.R.O.D.E. study—enlightens the potential role of intramuscular clodronate (CLO) as an analgesic drug in active painful EHOA as it seems able to obtain pain relief and better hand function although CLO does not modify the number of joint deformities [51,52].

There is still lack of evidence on efficacy of disease-modifying drugs as none of them were proven to prevent the progression of structural remodeling, which is the critical aspect of EHOA. Overall, the pharmacological therapeutic options are often inadequate; the lack of disease-modifying drugs lead a considerable amount of patients to face destructive lesions and consequent high functional impairment. On the other hand, there are several synovitis PsA treatment options, including disease-modifying antirheumatic drugs (DMARDs); among them Methotrexate has been identified as the drug of choice [53,54] by EULAR and GRAPPA recommendations.

New therapeutic drugs to treat refractory arthritis include biological agents such as TNF inhibitors (TNFi), IL-17 inhibitors (IL-17i), IL-12/23 inhibitor (IL-12/23i) and new targeted therapies such as a phosphodiesterase-4 inhibitor and a Janus kinase (JAK)/signal transducer and activator of transcription (STAT) inhibitor.

However, PsA distal interphalangeal involvement synovitis was not evaluated as a separate domain in any trials so data on this feature are lacking.

## 2. Role of Conservative Therapy

### 2.1. EHOA

International guidelines recommended to combine pharmacological and non-pharmacological modalities in the management of hand osteoarthritis [39,43].

The purpose of rehabilitation is to recover functional hand use and improve the pain.

A recent systematic review identified 28 studies debating the most relevant non-pharmacological therapeutically strategies for hand osteoarthritis [39].

They include a wide range of different interventions: joint protection program, use of splint and orthoses, range of motion and strengthening exercise, hand exercise with electromagnetic therapy, application of heat with paraffin wax or balneotherapy [55].

Most research studies combined different interventions in terms of type, duration and intensity/frequency but there is no a shared program of conservative non-pharmacological intervention.

#### 2.1.1. Finger Exercise Program

The technical expert panel of American College of Rheumatology (ACR) and the Arthritis Foundation provide strong recommendations to use exercises in treatment of HAO [37,43].

Hand exercise was demonstrated to have some beneficial effects on hand pain, function and finger joint stiffness [56,57]. As pain is one of the major causes for functional impairment, exercise thus could promote independence in day-by-day activity and also participation in social and working activity.

Finger exercise programs usually include strengthening of hand joint stabilizer muscle aiming to improve joint stability, active and passive range of motion exercises to prevent ankylosis and reduce pain and strengthening exercises with the purpose to enhance grip strength and dextrality [39,56,58,59].

Accumulating evidence showed that exercises are well tolerated among patients [57]; however, passive joint mobilization is operator dependent.

Beasley et al. recently reviewed three studies supporting the role of resistive exercise to improve grip strength [60,61,62].

In a more recent study Kang et al. compared finger exercise plus paraffin wax with paraffin wax alone. The author observed significant increased grip strength in patients who underwent finger exercise compared to those who received paraffin bath therapy alone [63].

In another RCT, Stoffer-Marx et al. proved the effect of combined intervention (education plus home exercise program) on hand strength in patients with HOA [64].

However, there is no accordance in literature about the role of exercise on the strength improvement in HOA. Furthermore, in a RCT Østeras et al. observed that exercises had beneficial improvement in short-term follow-up in terms of hand pain relief and reduction of stiffness but patients did not acquire any improvement in grip strength and hand dexterity [57].

The reinforcement of hand grip and pinch is the central part of hand rehabilitation. However excessive exercise could also stress painful joints [65]. For this reason, hand aROM (active range of motion) and resistive exercise should always be combined with joint protection or be executed avoiding discomfort for the patients and focusing on the principle of respecting the pain. The RCT by Dziedzic et al. demonstrated a significant improvement in pain with a combination of joint protection education and hand aROM and strengthening exercises [58,66].

Duration, intensity and frequency as well as the type of exercise regimen vary largely in different studies. This heterogeneity made it impossible to prescribe specific exercise programs and may be the reason for the lack of strong shared recommendation [43,55,56].

#### 2.1.2. Electromagnetic Therapy

Electromagnetic therapy has been investigated as an adjuvant therapy to treat hand OA [67]. In a randomized controlled single-blind follow-up study Kanat et al. examined the effect of magnetotherapy and hand exercises. These combined interventions were found to achieve a decrease in pain and stiffness, though enhanced hand function and quality of life [67].

Pulsed magnetic fields are especially indicated for muscular-skeletal tissue for their ability to penetrate deeply in the tissue thus having the possibility of stimulating the repair process. The mechanism is based on the flow of ionic current caused by electrical charges that stimulates growth factors synthesis and takes part in the resolution of inflammatory process through the activation of adenosine receptors [68].

#### 2.1.3. Paraffin Wax and Balneotherapy

Paraffin wax and other thermal modalities have been proven to decrease joint pain 12 weeks after treatment [55,69,70].

DileK et al. demonstrated in a single-blinded RCT on hand OA that paraffin bath therapy was effective in reducing pain and tenderness and in maintaining muscle strength up to 12 weeks follow-up.

Paraffin wax therapy uses local application of heat to obtain the vasodilation producing hyperemia, increasing transduction of tissue fluid and increasing lymph flow and the absorption of exudates [69]. Furthermore, heat produces analgesia and relaxes muscle tone; the therapeutic effects result in pain relief and reduction of muscular spasm [71].

Paraffin wax was combined with other types of intervention (hand exercise, home exercise program and education) with significant improvement in functional status and reducing pain [72,73].

Balneotherapy is another thermal modality having silver level of evidence in OA [55]; the application of mineral water had analgesic effects and increased quality of life. Recent research conducted by Horvat et al. demonstrated a significant improvement in hand function and pinch strength through balneotherapy combined with magnetotherapy [70].

#### 2.1.4. Occupational Therapy and Education

The central aspect of rehabilitative treatment is the education of patients to spare affected articulations and reduce pain during day-to-day activity.

In a RCT conducted by Dziedzic et al. it has been shown that occupation therapy can support self-management in older adults with hand osteoarthritis [58].

Occupational therapy is the rehabilitation discipline that aims to restore or maintain the independence in the execution of basic activity of daily living (A.D.L.) as well as more complex ones (I.A.D.L.). Exercises of OT (Occupational therapy) for interphalangeal joints respect the following principles: promote joint movements in the most stable anatomical plan, favor bi-manual grasp, avoid repetitive pinch grasping and twitch movement. To this extent, it is possible to cut down on forces stressing the affected joints, reducing inflammation and strain on soft tissue and preventing damage of non-affected joint surfaces [58,74].

Pain relief and everyday life activity could be also promoted by educational programs to self-management strategies, such as activity adaptation or use of assistive devices [75].

Only one study assessed the effect of assistive technology (AT) to support activity of daily living. In a RCT Kjeken compared home exercise program to home exercise program with assistive device or splint in patients with HOA. AT was chosen depending on specific activity limitations reported by patients. The author demonstrated that AT is a useful strategy to improve activity performance [75].

In summary conservative therapies can support pharmacological and surgical options. Moreover, if drug therapies and surgery are not effective, they could be used as a stand-alone option to decrease pain and improve function as a treatment goal. Many authors suggested effective therapeutic regimens that differ in terms of modality, time and intensity. Therefore, it could be desirable in the future to reach an international consensus for the identification of standardized protocols.

### 2.2. Psoriatic Arthritis

Concerning conservative treatment strategies to treat PsA, current evidence is still weak.

The prolonged joint immobility due to pain and inflammation promotes stiffness, ankylosis and joint dislocation. To this extent, passive and active joint mobilization can help pharmacological therapy promoting joint mobility and interrupting the vicious circle of pain-immobility-rigidity. It is important to avoid, as for HEOA, muscular fatigue since it could perpetuate pain and enhance inflammation of joints [76].

However, international guidelines recognized the role of general physical therapy and exercise, but evidence about the specific protocols is lacking [54,77].

Group for Research and Assessment of Psoriasis and Psoriatic Arthritis (GRAPPA) supported the recommendation for physiotherapy for axial PsA or in case of patients’ refractory to bDMARDs, while they do not specifically address use of any non-pharmacological treatment for peripheral arthritis [77].

In the updated American College of Rheumatology/National Psoriasis Foundation guidelines general exercise and physical therapy intervention present low-to-very-low quality of evidence [78].

Moreover, only evidence of moderate quality is present supporting the role of resistive exercises in PsA patients [79].

EULAR published a recent update on physical activity for inflammatory arthritis that could be applied to psoriatic arthritis. They recommended physical activity as an integral part of standard care [80]. However, type and dosage of exercise and physical activity has not been assessed so far.

In summary, the value of non- pharmacological options in PsA currently seems of secondary importance with stronger evidence of efficacy of conservative therapies for HEOA.

## 3. Conclusions and Research Agenda

HEOA and PsA with DIP involvement are common diseases affecting the hand and having a progressive and painful evolution. Currently there are no effective drugs able to modify HEOA the course of the disease while corticosteroids and CLO or HQC are used only for short courses to improve symptoms. Furthermore, data on drug effectiveness of PsA with DIP involvement are lacking. Both diseases evolve with a progressive limitation in grip due to limited range of motion of the affected joints and stenosing tenosynovitis.

Conservative therapy should be considered in order to reduce pain and improve hand functionality. Further research is needed to find the correct place of physical therapy to prevent stiffness and ankylosis due to the vicious circle of inflammation-pain-immobility-rigidity.

## Figures and Tables

**Figure 1 jcm-10-02630-f001:**
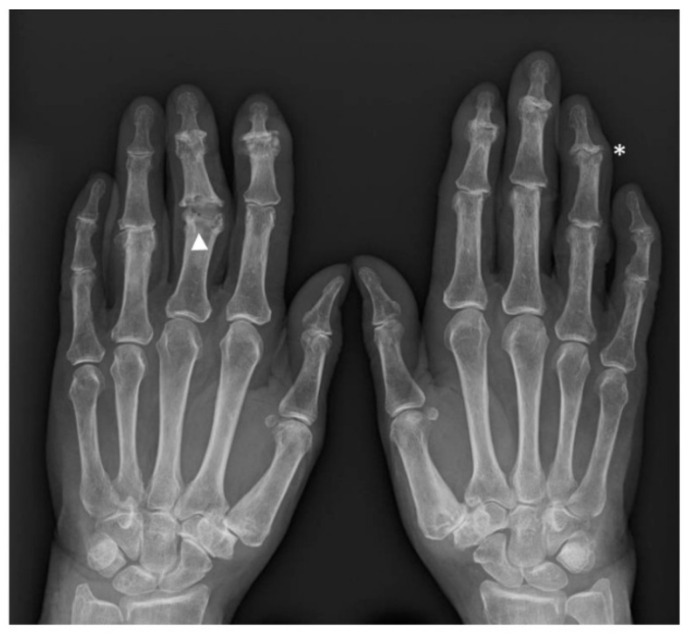
X-ray images of interphalangeal involvement in EHOA: gull wing appearance along with osteophytes and sclerosis (asterisk) and saw tooth appearance (triangle).

**Figure 2 jcm-10-02630-f002:**
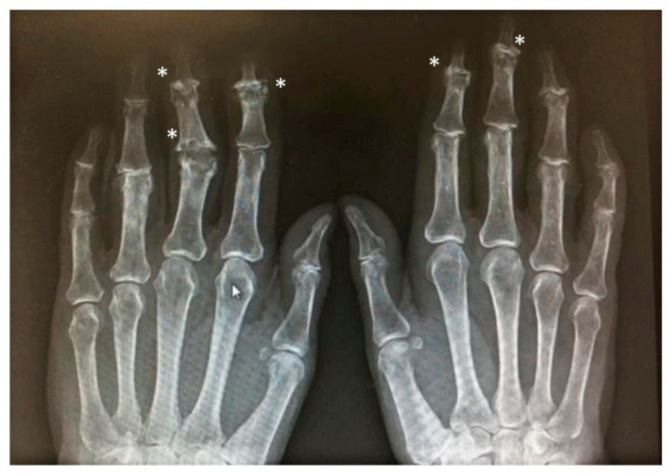
X-rays of hand show radiologic features of interphalangeal involvement in PsA (asterisk).

**Figure 3 jcm-10-02630-f003:**
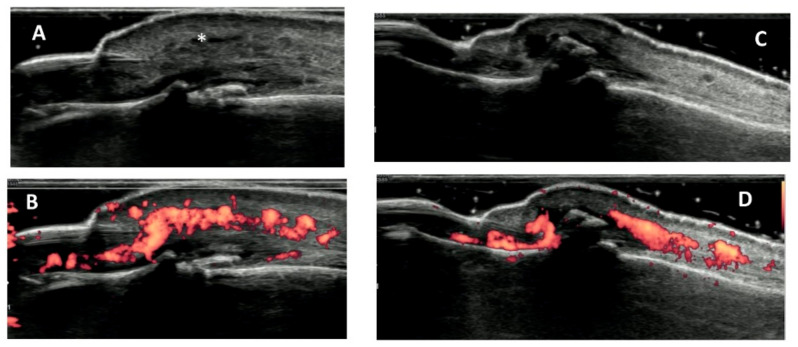
Ultrasound images of distal interphalangeal involvement in PsA ((**A**) gray scale; (**B**) power doppler) and OA ((**C**) gray scale; (**D**) power doppler). The asterix (*) highlights the soft tissue oedema surronding the extensor tendon in PsA.

**Table 1 jcm-10-02630-t001:** Demographic, clinical and imaging features of patients with Erosive Hand Osteoarthritis (HEOA) and Psoriatic Arthritis (PsA).

	HEOA	PsA
Demografic caracteristics	Middle-aged women	Middle-aged woman and man
Clinical features	- DIP and PIP joints involvement, with preference for DIP - Heberden’s nodes (DIP) and Bouchard’s nodes (PIP)	- typically DIP and PIP joints involvement, with preference for DIP - skin involvement - dactylitis - arthritis mutilans - spinal involvement - enthesis - nail dystrophy
RX features	- erosions (begin at the central portion of articulation, sharply marginated) - “gull wings” appearance - “Saw tooth” appearance - joint space narrowing - subchondral sclerosis - osteophytes, linear bone apportion (predominantly at the margins of DIP and PIP joint)	- erosions (begin at the marginal part of the joint and extend into the center) - “mouse ears” appearance - pencil in the cup” deformity - joint space narrowing - subchondral sclerosis - spike-like or fluffy proliferative alterations surrounded erosions on joint surface margins and phalangeal tuft - all joints of a digit involved—ray distribution
US features	- synovial thickening - osteophytes	- nail psoriasis - synovitis - enthesis - dactylitis

## Data Availability

Not applicable.

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
