# Peer review of "Hand Erosive Osteoarthritis and Distal Interphalangeal Involvement in Psoriatic Arthritis: The Place of Conservative Therapy"

_jcm, 2021, doi:10.3390/jcm10122630_

Round 1
Reviewer 1 Report
Thank you very much for your revision. In the first review, I suggested plain xray is helpful in daily practice. Please add figure 2 and/or 3 with typical PsA vs EOA plain xray films (you can use arrow/arrow-head for indicating important differences)Author Response
Point 1: Thank you very much for your revision. In the first review, I suggested plain xray is helpful in daily practice. Please add figure 2 and/or 3 with typical PsA vs EOA plain xray films (you can use arrow/arrow-head for indicating important differences)
Response: We thank the reviewer for this comment. In line 148 we added figures 2 and 3 (see the file below).

Reviewer 2 Report
Dear authors,
the manuscript has benefited greatly from your improvements. To further increase the quality, I recommend a critical review of the style (spaces, paragraphs, abbreviations, such as et. al.). In terms of content, it is now a successful work.
Author Response
Point 1: the manuscript has benefited greatly from your improvements. To further increase the quality, I recommend a critical review of the style (spaces, paragraphs, abbreviations, such as et. al.). In terms of content, it is now a successful work.
Response: We thank the reviewer for this comment. We have made the required structural changes to the text.
Round 2
Reviewer 1 Report
Manuscript has been much improved. One more final comment to consider. Please add a reference as below for MRI of PsA patients. Magnetic Resonance Angiography in Psoriatic Arthritis of the Hand RYO ROKUTANDA, MITSUMASA KISHIMOTO and MASATO OKADA The Journal of Rheumatology August 2012, 39 (8) 1700This manuscript is a resubmission of an earlier submission. The following is a list of the peer review reports and author responses from that submission.
Round 1
Reviewer 1 Report
The authors of “Hand erosive osteoarthritis and distal interphalangeal involvement in Psoriatic Arthritis: the place of conservative therapy” compare two diseases, hand erosive osteoarthritis (HEOA) and psoriatic arthritis (PsA), both of which typically have manifestations at the DIP joints. An overview is given of the pathogenesis of both diseases, the clinical manifestations and the therapy. Therapeutic options are discussed in terms of medications as well as surgical and conservative options. Finally, the authors suggest that conservative therapy should be considered as a treatment option in both diseases.
A strength of the manuscript is that the various aspects of HEOA and PsA in terms of their clinical manifestation or in terms of conservative therapeutic options are mentioned in a very comprehensive and detailed manner.
The authors title their abstract "the place of conservative therapy" and they provide a good overview of the conservative therapy options and also the non-conservative options. However, the reader expects an assessment of the extent to which conservative therapies are sufficiently used, whether they should be applied more intensively, and how they should be considered in relation to the non-conservative therapy options, i.e., in accordance with the current literature, should conservative therapy be applied as an accompanying option to medical and surgical therapy or as the stand-alone option?
It would also be desirable to assess, on the basis of the evidence presented, the extent to which the value of conservative therapy is the same or possibly different in HEOA and PsA.
The image diagnostic features for HEOA and PsA are accurately listed, especially for x-rays. In addition, the diagnostic possibility of ultrasonography is discussed. Especially for short-term developing lesions (synovialitis, cartilage degeneration, bone marrow oedema) the MRI technique has proven to be increasingly helpful in recent years. It would therefore be appropriate to add MRI diagnostics to the range of diagnostic possibilities.
Please be more clear in your statement about the therapy option of PsA. On the one hand you write that there is no clear disease modifying therapy (line 168) and on the other hand you quote EULAR that there are several therapy options with DMARDs (line 200).
Minor Notes:
Line 18: consider – considered
Line 22: heath – heat
Line 31: Please introduce abbreviations the first time – PsA (and others)
Line 63: Please add reference.
Line 70: challenge – challenges
Line 79/81: present or past? Choose a time in which you write and stay in it as far as possible.
Line 93: tough - though
Line 106: Please add reference.
Line 123: Please add reference and consider the relevance of MRI in PsA (for example doi: 10.1186/ar1934).
Line 128: Please add reference.
Line 138: Osteolytic lesion AND bone proliferation… Especially PsA is characterized by bone proliferation next to osteolytic lesions
Line 150: EOA – EHOA
Line 182: Please introduce abbreviations the first time – RCT
Author Response
Point 1: The authors title their abstract "the place of conservative therapy" and they provide a good overview of the conservative therapy options and also the non-conservative options. However, the reader expects an assessment of the extent to which conservative therapies are sufficiently used, whether they should be applied more intensively, and how they should be considered in relation to the non-conservative therapy options, i.e., in accordance with the current literature, should conservative therapy be applied as an accompanying option to medical and surgical therapy or as the stand-alone option?
Response 1: We thank the reviewer for this comment. In line 331 we added the next sentence: “In summary conservative therapies can support pharmacological and surgical options. Moreover, if drug therapies and surgery are not effective, they could be used as a stand-alone option to decrease pain and improve function as a treatment goal. Many authors suggested effective therapeutic regimens that differ in terms of modality, time and intensity. Therefore, it could be desirable in the future to reach an international consensus for the identification of standardized protocols.”
Point 2: It would also be desirable to assess, on the basis of the evidence presented, the extent to which the value of conservative therapy is the same or possibly different in HEOA and PsA.
Response 2: We thank the reviewer for this comment. To better clarify this concept, in line 349 we added this sentence: “In summary, the value of non- pharmacological options in PsA currently seems of secondary importance with stronger evidences of efficacy of conservative therapies for HEOA”.
Point 3: The image diagnostic features for HEOA and PsA are accurately listed, especially for x-rays. In addition, the diagnostic possibility of ultrasonography is discussed. Especially for short-term developing lesions (synovialitis, cartilage degeneration, bone marrow oedema) the MRI technique has proven to be increasingly helpful in recent years. It would therefore be appropriate to add MRI diagnostics to the range of diagnostic possibilities.
Response 3: Thank for highlighting this significant point. Certainly MRI could represent a useful tool to better detect inflammatory lesions and bone erosions. In line 179 we added this sentence “Further imaging modalities provide reliable results in detection of typical lesions of both HEOA and PsA and help to make accurate diagnosis, as proved in clinical trials [Filippou]. As mentioned, differential diagnosis between HEOA and PsA may sometimes become largely dependent on imaging interpretation results. Magnetic resonance imaging (MRI) is a valid operator-independent tool and it has a higher sensitivity than radiography and US in detecting bone erosion, especially in earlier stages of HEOA [Grainger 2007, Colebacth-EULAR 2013]. Moreover, MRI can detect damage in all the joints structures in PsA patients (i.e. cartilage defects, bone marrow oedema, soft tissue oedema and synovitis) [McQueen 2006]. MRI is not mandatory in all patients but it could support diagnosis when clinical and other radiological findings are not conclusive.”
Point 4: Please be more clear in your statement about the therapy option of PsA. On the one hand you write that there is no clear disease modifying therapy (line 168) and on the other hand you quote EULAR that there are several therapy options with DMARDs (line 200).
Response 4: We thank the reviewer for this comment. As you kindly suggest the current treatment for PsA include different options with DMARDs . We modifing the sentence in line 188-189 as following: “Pharmacological options currently available for the treatment of EHOA are mostly symptomatic with no clear improvement on diseases’ progression to erosive destructive lesions”
Reviewer 2 Report
It is my great pleasure to review this manuscript titled "Hand erosive osteoarthritis and distal interphalangeal involvement in Psoriatic Arthritis: the place of conservative therapy". The manuscript is well written, I have a major and minor comments to this manuscript.
Major comments:
In the background section, you might consider adding more details on the differential diagnosis in the daily practice setting, eg Mody E, et al. Br J Dermatol, 2007, 1050-1051. Kishimoto M, et al. Best Pract Res Clin Rheumatol, online published on Mar 17, 2021).
It is also helpful to have tables and figures, Erosive OA vs PsA. I would like to suggest one table regarding clinical features, age and demographics, imaging including US and Plain Radiograph.
Figures are plain xrays typical EOA vs PsA and you can point out what is the difference between the two.
In the recent advancement of imaging technique, several authors pointed out the EOA might be co-incident with CPPD or crystal induced inflammatory arthritis. You may add these references as well, especially using Dual energy CT to pick up crystal lesions in the patient with OA.
Minor comments
In the study for EOA, I believe that there is a positive study using IL1 inhibitor for erosive OA (Knee?). Please add the reference and comment in the text.
Author Response
Point 1: In the background section, you might consider adding more details on the differential diagnosis in the daily practice setting, eg Mody E, et al. Br J Dermatol, 2007, 1050-1051. Kishimoto M, et al. Best Pract Res Clin Rheumatol, online published on Mar 17, 2021).
Response 1: We thank the reviewer for this comment. In line 90 we added the next sentence. “Thus, a prompt diagnosis of PsA can be achieve with an accurate clinical assessement and evaluation of skin and nail lesions in patients presenting with other musculoskeletal disorders [Kishimoto 2021]. With this purpose, in clinical setting the presence of a multidiscliplinary team including dermatology and rheumatology specialists could increase early diagnosis and optimize care of PsA patients [Mody 2007].”
Point 2: It is also helpful to have tables and figures, Erosive OA vs PsA. I would like to suggest one table regarding clinical features, age and demographics, imaging including US and Plain Radiograph.
Response 2: We thank the reviewer for this comment. We added in line 74 table 1 (please see the attachment)
Point 3: Figures are plain xrays typical EOA vs PsA and you can point out what is the difference between the two.
Response 3: We thank the reviewer for this advice. In line 176 we added figure 1 (please see the attachment)
Point 4: In the recent advancement of imaging technique, several authors pointed out the EOA might be co-incident with CPPD or crystal induced inflammatory arthritis. You may add these references as well, especially using Dual energy CT to pick up crystal lesions in the patient with OA.
Response 4: Thank you for your kind advice. In line 188 we added this sentence “It has been shown that PsA and HEOA could coexist with gout and calcium pyrophosphate deposition disease (CPPD) [2, Filippou 2020]. In particular, considering HEOA, joint damage has a crumbling appearance similar to CPPD and CPP deposition are not so rarely detected in osteoarthritis; thus differential diagnosis could be difficult. Recently, it has been demonstrated that Dual energy Computerized Tomography (DECT) directly visualizes and quantifies calcium pyrophosphate (CPP) and urate monosodium (MSU) crystal [Filippou 2020]. Moreover, DECT, in contrast with conventional CT, DECT is able to detect inflammatory lesions. Hence in this context DECT seems a valuable tool to achieve a more definite diagnosis.”
Point 5: In the study for EOA, I believe that there is a positive study using IL1 inhibitor for erosive OA (Knee?). Please add the reference and comment in the text.
Response 5: We thank the reviewer for this comment. IL-1 inhibitors injections has been evaluated in knee OA [Martel-Pelletier J, Pelletier JP. Osteoarthritis: A single injection of anakinra for treating knee OA? Nat Rev Rheumatol. 2009 Jul;5(7):363-4. / Chevalier X, Goupille P, Beaulieu AD, Burch FX, Bensen WG, Conrozier T, Loeuille D, Kivitz AJ, Silver D, Appleton BE. Intraarticular injection of anakinra in osteoarthritis of the knee: a multicenter, randomized, double-blind, placebo-controlled study. Arthritis Rheum. 2009 Mar 15;61(3):344-52]. But we decided not included this data as this review focuses on efficacy of different therapeutical strategies for erosive hand osteoarthritis and no strong recommendation emerged for IL-1 inhibitors within this purpose.”
